

# Identification of genetic loci for early maturity in spring bread wheat using the association analysis and gene dissection

Antonina A. Kiseleva[1,2], Irina N. Leonova[1], Elena V. Ageeva[3],
Ivan E. Likhenko[3] and Elena A. Salina[1,2]

[1] Laboratory of Plant Molecular Genetics and Cytogenetics, The Federal State Budgetary
Institution of Science Federal Research Center Institute of Cytology and Genetics, Siberian
Branch of the Russian Academy of Sciences, Novosibirsk, Russia
[2] Kurchatov Genomics Center, Institute of Cytology and Genetics SB RAS, Novosibirsk, Russia
[3] Laboratory of Field Crop Breeding and Seed Industry, Siberian Research Institute of Plant
Production and Breeding, Branch of the Institute of Cytology and Genetics, Siberian Branch of
the Russian Academy of Sciences, Novosibirsk, Russia

## ABSTRACT

**Background:** Early maturity in spring bread wheat is highly desirable in the regions
where it enables the plants to evade high temperatures and plant pathogens at the end
of the growing season.
**Methods:** To reveal the genetic loci responsible for the maturity time association
analysis was carried out based on phenotyping for an 11-year period and
high-throughput SNP genotyping of a panel of the varieties contrasting for this trait.
The expression of candidate genes was verified using qPCR. The association between
the SNP markers and the trait was validated using the biparental $F_{2:3}$ population.
**Results:** Our data showed that under long-day conditions, the period from seedling
to maturity is mostly influenced by the time from heading to maturity, rather than
the heading time. The QTLs associated with the trait were located on 2A, 3B, 4A, 5B,
7A and 7B chromosomes with the 7BL locus being the most significant and
promising for its SNPs accelerated the maturity time by about 9 days. Gene dissection
in this locus detected a number of candidates, the best being *TraesCS7B02G391800*
(bZIP9) and *TraesCS7B02G412200* (photosystem II reaction center). The two genes
are predominantly expressed in the flag leaf while flowering. The effect of the SNPs
was verified in $F_{2:3}$ population and confirmed the association of the 4A, 5B and 7BL
loci with the maturity time.

# INTRODUCTION

Spring common wheat is a grain crop whose cultivation is important for many countries.
In the Russian Federation, the main territories growing spring wheat include the Volga,
Southern Urals, Altai, Western and Eastern Siberia regions. These regions are located in
the climatic zones having different daylength, rainfall and the sum of effective
temperatures. Despite these differences, most of them are characterized by sudden changes

Corresponding author
Antonina A. Kiseleva,
antkiseleva@bionet.nsc.ru

in temperature and rainfall during sowing and harvesting, which may affect the productivity.

The duration of the growing season and phenological developmental phases are important factors affecting the expression of agronomically important traits of common wheat. The duration of heading date and maturity period has been shown to affect such important traits as yield, grain protein content, protein accumulation and composition in wheat and barley (*Johansson et al., 2005*; *Johansson, Prieto-Linde & Gissén, 2008*; *Malik, 2012*). One of the significant problems Russian breeders face is cultivation of highly adaptive early-maturing wheat varieties since accelerated maturity enables one to vary the time of sowing and harvesting, avoiding unfavorable environmental conditions. Earliness may also be helpful not only in evading the negative climatic factors but also in preventing different biotic stresses (*Poehlman & Sleper, 1995*), which makes detecting novel sources of earliness an actual scientific problem.

The main genetic factors affecting the vegetative period length, and eventually development duration are vernalization, photoperiod sensitivity gene and earliness loci. *Kamran et al. (2013)* have demonstrated that the *Ppd-D1a* allele can affect not only the flowering time but also may speed up the maturity time in wheat. Moreover, several studies have identified a locus on the short arm of the 2D chromosome that is associated with the maturity time (*Prasad et al., 2003*; *Chen et al., 2015*; *Perez-Lara et al., 2016*; *Zou et al., 2017*).

The position of this locus on the chromosome coincides with the localization of the *Ppd-D1* gene to be another argument in favor of the fact that this gene can participate in determining not only the time of heading and flowering but also the maturity time. *McCartney et al. (2005)* have shown that the 4A maturity time locus is associated with the *Wx-B1* gene, but supposed that the locus should be determined by the genes located nearby *Wx-B1* and not by the *Waxy* gene itself.

Other studies have demonstrated that the wildtype allele of *NAM-B1* (6B) gene might cause faster maturity (*Hagenblad et al., 2012*). However, it has been observed that different combinations of *NAM-A1* and *NAM-B1* alleles have different effects on wheat maturity time. Thus, varieties with *NAM-A1a* and *NAM-A1b* alleles were early and medium-early maturing, regardless of their *NAM-B1* allelic composition. At the same time, varieties with *NAM-A1c* and *NAM-A1d* alleles were middle and late maturing only in combination with a non-functional *NAM-B1* allele. Conversely, the *NAM-A1c* and *NAM-A1d* alleles combined with functional/deletion *NAM-B1* alleles conferred early and mid-early maturity (*Alhabbar et al., 2018*).

A number of studies involving genetic mapping and analysis of the quantitative trait loci (QTL) responsible for maturity time have demonstrated that the loci associated with this trait are detected in the most wheat chromosomes: 1A, 1B, 1D, 2A, 2B, 2D, 3B, 3D, 4A, 4B, 4D, 5A, 5B, 5D, 6B, 6D, 7A, 7B, and 7D (*Kulwal et al., 2003*; *McCartney et al., 2005*; *Huang et al., 2006*; *Wang et al., 2009*; *Kamran et al., 2013*; *Chen et al., 2015*; *Perez-Lara et al., 2016*; *Zou et al., 2017*).

Maturity time has been extensively studied in some other plant crops, such as barley, soybean, and rice. In barley, loci associated with maturity time are found on all the

chromosomes (*Saade et al., 2016*). The most significant QTLs are related to the 2H, 5H, and 7H chromosomes. One of the barley genes suggested to be involved in maturity is *HvELF3*. The wild allele of this gene causes earlier flowering and maturity and increases the harvest index (*Saade et al., 2016*). ELF3 was identified in wheat and its association with heading time was established in previous studies (*Wang et al., 2016*; *Alvarez et al., 2016*, *2023*; *Wittern et al., 2023*). However, its impact on maturity time has not been evaluated. In a more recent study by *Wang et al. (2023)*, they suggested that *TaELF3* may be a candidate for early maturity, but these conclusions were based solely on previous studies that demonstrated its effect on heading time.

The *Denso* locus (*sdw1*) on the 3H chromosome has also been shown to accelerate maturity (*Jing & Wanxia, 2003*; *Saade et al., 2016*). *Gibberellic acid (GA)-20 oxidase* has been proposed as a candidate for the *sdw1* locus (*Jia et al., 2009*). The orthologs of the barley sdw1 locus gene, *TaGA20ox1*, which is involved in the GA pathway, have been previously mapped in wheat (*Appleford et al., 2006*). However, the specific traits affected by these genes have not been estimated. *Chen et al. (2016)* demonstrated that another gene in the GA pathway, *Rht1*, is associated with maturity time itself and, in combination with *Vrn-B1* and *Ppd-D1* genes, determines sensitivity to this phytohormone. In contrast, *Vrn-B1* and *Ppd-D1* alone did not show an effect on maturity time in that study.

Investigation of soybean maturity time has revealed the loci related to the trait on chromosomes 1, 11, 14, and 19 (*Hu et al., 2020*). There are nine loci in soybean known to affect both heading and maturity time (*Watanabe et al., 2009*). A more detailed study of these loci has identified genes associated with the studied traits. These are the *GmFT2a* and *GmFT5a* genes (*Flowering Locus T* homologue of Arabidopsis); the *GIGANTEA* gene homologue regulating *CONSTANS* and *FT* expression; and the *GmPHYA3* and *GmPHYA2* phytochrome genes (*Liu et al., 2008*; *Watanabe, Harada & Abe, 2011*; *Xia et al., 2012*). The genes *FT*, *GIGANTEA*, and *CONSTANS* in wheat have been identified as regulators of heading and flowering time (*Yan et al., 2006*; *Shaw et al., 2020*; *Li et al., 2023*). However, their involvement in regulating maturity time remains unexplored. While these genes are conserved in various crops, including wheat, their specific roles in different species may vary (*Watanabe et al., 2011*). Hence, it is crucial to investigate whether these genes play role in determining maturity time precisely in wheat.

Despite the large amount of available data, the genes known to be associated with maturity time in other species (except *NAM1*) have not yet been concerned with maturity in wheat, so in the present study a collection of spring wheat varieties has been investigated for their heading and maturity time to identify the putative genetic loci responsible for the maturity trait.

## MATERIALS AND METHODS

### Plant material and phenotyping

In this study a collection of 92 Russian spring wheat varieties was investigated that originated from the breeding centers of seven regions such as Samara, Altai, Novosibirsk, Krasnoyarsk, Kemerovo, Tyumen, and Omsk. The varieties were maintained in the GeneAgro seed bank of Institute of Cytology and Genetics of Siberian Branch of the

Russian Academy of Sciences (ICG SB RAS) (https://ckp.icgen.ru/plants/). The wheat varieties had been adapted to the climatic conditions of Western Siberia whose summer period is rather short, so different environmental troubles often happen at the end of the vegetation period. More detailed information about the wheat varieties' origin is presented in *Leonova et al. (2022)*.

For mapping the genetic loci, we obtained two phenotyping datasets for this panel of spring wheat varieties. The first dataset was obtained in 2018 and described the heading, heading to maturity and maturity periods to cover the total seedling to maturity time. The second dataset included 10-year data (from 2005 to 2014) evaluating the time from seedling to maturity.

The plants were cultivated in the two experimental fields of ICG SB RAS in the Novosibirsk Region (field 1: 54.884987°N, 82.899519°E; field 2: 54.914070°N, 82.975379°E). The accessions were phenotyped on both fields in 2018 and on field 1 in 2005–2014.

The phenological stages were evaluated following the Zadoks classification (*Zadoks, Chang & Konzak, 1974*). The heading time was recorded at the moment when 70% of the plants in the plot had emerged with the ear from the tube by one-half, and it was calculated as the number of days from seedling to heading (corresponding to stage GS55).
The maturity was determined by hard grains, yellowing, and drying of the ears and stems (corresponding to stage GS92).

The seeds were sown in the second half of May in 2 repetitions on plots of 2 m$^2$. The soil in both fields was leached chernozem. The humus thickness varied from 40 to 60 cm, and the humus content was 4.2%. The soil was slightly acidic (pH 6.7–6.8), with its nitrogen content being 0.34%, total phosphorus-0.30% and potassium-0.13%.

To test the effect of identified SNPs, an $F_{2:3}$ population was obtained from a cross of mid-season cultivar Obskaya 2 and early-maturing cultivar Novosibirskaya 15.
The population was phenotyped in 2018 in field 1 in 2 repetitions on plots of 0.25 m$^2$.

The weather data (average daily temperature and humidity by months) provided by a meteorological station located in the Novosibirsk region at 54.90 N, 82.95 E and an altitude of 131 m were taken from http://www.pogodaiklimat.ru. The air temperature and precipitation amount during the experiments were contrasting both during one growing season and over the years. The most optimal weather to ensure high productivity was registered in 2005 and 2009, whose temperatures and precipitation amounts in these years were closest to the mean seasonal values. The moisture supply was insufficient in 2008 (166.8 mm), 2010 (135.9 mm), 2011 (153.5 mm), 2012 (102.7 mm) and 2014 (176 mm). An extremely severe drought was noted in July 2012, with 3.7 mm of rain falling during the month, which was only 6.1% of the mean seasonal value.

Growing degree days (GDD) from seedling to maturity, were calculated according to the formula of *McMaster (1997b)*: GDD = (Tmax + Tmin)/2 − Tbase, where Tmax and Tmin are the maximum and minimum temperatures for the day and Tbase was 0 °C (*McMaster, 1997a*). The GDD was highest in 2012 (more than 2,200). Cold and rainy weather at the beginning of the growing season was observed in 2007, 2010, 2013, 2014 and 2018 with GDD less than 1,600. The most optimal air temperature for plant development was in 2008

and 2014 when the GDD varied from 1,800 °C to 1,900 °C. The weather was especially unfavourable for seedlings in 2018 when the average air temperature was 4 °C below the norm in May and the precipitation exceeded the mean seasonal value by 126.8%.

## DNA extraction and high-throughput genotyping

The DNA was extracted from young leaves using the modified sodium bisulfite protocol described by *Kiseleva et al. (2016)* and purified using the Bio-Silica Kit for DNA Purification from Reaction Mixtures as per the manufacturer's protocol. The samples were genotyped using the Illumina Infinium 15 K Wheat containing 13,007 SNPs by TraitGenetics GmbH (www.traitgenetics.com).

## Statistical analysis

Analysis of variance (ANOVA) to explore the association between traits was performed using the 'aov' function in R. Spearman's correlation coefficients ($r^2$) were calculated. The model considering genotypic effects as fixed and environmental effects as random was used to calculate the best linear unbiased estimates (BLUEs) of each genotype. To check if the distribution of the traits was normal, we used the Shapiro–Wilk normality test (R function). To estimate the maturity and heading time heritability, all effects except the intercept were assumed as random, so the component variance was estimated using R package lme4 (*Bates et al., 2020*) and $H^2$ was calculated using the "Standard" broad-sense heritability method (*Holland, Nyquist & Cervantes-Martínez, 2003*).

## Association analysis

Only polymorphic markers (SNPs) were selected for subsequent analysis. The genotyping data were filtered by minor allele frequency (MAF > 0.05) and by missing data less than 5% of population. Population structure analysis was performed using the LEA algorithm (snmf function with K = 1–10) (*Frichot & François, 2015*).

A mixed linear model (MLM) was performed in TASSEL V5.1 to carry out the association analysis, which accounted for fixed effects of SNPs and population structure (Q) and random effects for kinship (K). Quantile-quantile and Manhattan plots were generated using the R package "GWASTools" (*Conomos et al., 2012*).

Linkage disequilibrium (LD) between SNP markers was calculated using the R package "genetics" (*Warnes et al., 2019*). LD decay plots were generated using R package "LDheatmap" (*Shin et al., 2006*). The chromosome positions of SNP markers were established according to the IWGSC Reference Sequence v1.0 Annotation (*Consortium IWGS, 2018*).

## Annotation of candidate genes

To provide insight into the functional specification of candidate genes, they were annotated using the IWGSC 1.0 data. To assess the gene expression, the developmental time course of the common wheat cultivar Azhurnaya (*Ramírez-González et al., 2018*) deposited in Wheat Expression Browser expVIP (*Borrill, Ramirez-Gonzalez & Uauy, 2016*) was used. Visualization and clustering were performed using R package d3heatmap.

Gene-specific primers were designed in PrimerQuest Tool software (Integrated DNA Technologies, Inc., Coralville, IA, USA) (Table S1).

## RNA isolation and RT-qPCR analysis

According to the results of the association study, 10 wheat varieties with different haplotypes in the QTL located on 7B chromosome were selected. The varieties considered as early and late maturing and having the respective CCGACTGACT and TAAGTGAGCT haplotypes were cultivated in the year 2020 on the experimental field of ICG SB RAS in Novosibirsk. The plants were harvested at the start of the anthesis stage GS61 (*Zadoks, Chang & Konzak, 1974*). The stage was chosen based of the candidate gene expression data from Wheat Expression Browser expVIP (*Borrill, Ramirez-Gonzalez & Uauy, 2016*). Three replicates from each genotype were harvested into liquid nitrogen in 7 h after the beginning of the light period (about 10 a.m.). Each biological replicate consisted of three plants of each cultivar combined into a bulk, with each combination replicated three times (biological replicates) for a total of 30 plants sampled.

The RNA was extracted using Plant RNA MiniPrep (Zymo Research, Tustin, CA, USA), followed by DNase treatment with a RNase-Free DNase set (QIAGEN, Hilden, Germany). The quantity and quality of the extracted RNA were assessed using a QUBIT 4 fluorometer with RNA BR and RNA IQ kits respectively (Thermo Fisher Scientific, Vilnius, Lithuania). cDNA was synthesized using RevertAid First Strand cDNA Synthesis (Thermo Fisher Scientific, Vilnius, Lithuania) following the manufacturer's protocol with 2 µg of total RNA as a template and random hexamer primers (Thermo Fisher Scientific, Vilnius, Lithuania). In total, 2 µl of 20-fold final cDNA dilution was used for the following analysis.

The fluorescence data were collected using a QuantStudio™ 5 Real-Time PCR System (Thermo Fisher Scientific, Vilnius, Lithuania) with PowerUp SYBR Green Master Mix (Thermo Fisher Scientific, Vilnius, Lithuania). The measurements were performed in three technical replicates. The reaction products were checked using melting curve analysis.

To detect the most applicable reference genes, the expression of seven candidate reference genes selected according to the literature (Table S2) was analyzed. Applying the geNorm algorithm (*Vandesompele et al., 2002*) integared in qBase+ software (*Hellemans et al., 2008*) two genes (MetAP1 and SAR) were chosen as the best references (geNorm V < 0.15).

The PCR efficiencies were determined using the LinReg software (*Ruijter, Ilgun & Gunst, 2014*). The Calibrated Normalized Relative quantities (CNRQ) were calculated in qBase+ (*Hellemans et al., 2008*). The target-gene expression was normalized against two reference genes (MetAP1 and SAR). The plots representing expression ratios were generated with the ggplot2 (*Wickham, 2016*) and ggpubr (*Kassambara, 2023*) R packages.

## RESULTS

### Phenotyping

The 10-year period data included information only on the days from seedling to maturity (maturity time). To reveal correlations between the heading and maturity phenological phases, the times from seedling to heading (heading time, HT), from heading to maturity

(HTMT), and the total time from seedling to maturity (maturity time, MT) were evaluated for two environmental conditions in 2018. Using these data, HT and HTMT contributions to MT were estimated.

The results of this experiment demonstrated that the time from seedling to heading (HT) of the studied varieties ranged from 35 to 44 days, with a heritability of 53% for this trait. While the total time from seedling to maturity (MT) varied from 77 to 104 days (Fig. 1). The large MT variation and extension (about a 4-week difference) may be due to unfavorable weather conditions in 2018, when the temperature in May was 4 °C below the norm, and the precipitation during growing season significantly exceeded the average.

Correlation analysis of these parameters demonstrated that HTMT strongly correlated with MT (r = 0.96, *p*-value < 2.2E−16), while HT moderately correlated with MT (r = 0.58, *p*-value = 2.089E−10) (Fig. 1).

Ten-year evaluation (2005–2014) of MT duration showed that it varied from 60 to 96 days in different years (Table S3). ANOVA indicated that such parameters as the genetic component and environment both significantly contributed to MT (Table S4). Individual variance component analysis of the ten-year data demonstrated that the genotypic variance principally contributed to MT (73.5%).

Five varieties (Polushko, Novosibirskaya-15, Salimovka, Novosibirskaya 22, Lutescense 25) had the earliest maturity time (MT < 72 days) if compared with cv. Novosibirskaya 81, Omskaya 28, Tulaikovskaya zolotistaya, Tulaikovskaya 10, and Baganskaya 93 which matured later than the others (MT > 81 days). These varieties were subsequently used to study the expression of candidate genes as contrasting genotypes.

For further analysis and minimization in assessing the environmental impact, the best linear unbiased estimates based on the data for 11 years of observation were estimated (2005–2014, 2018).

## Association analysis

Association mapping for MT was performed using the BLUEs of 11-year phenotype estimation of 96 wheat varieties and its results are summarized in Table 1. A total of 22 significant SNPs were located on chromosomes 2A, 3B, 4A, 5B, 7A, and 7B (Fig. 2A, Table 1). Most of these chromosomes are represented by one or two significant markers except for chromosome 7BL where ten SNPs were tightly linked to each other (Fig. 2C, Table 1). In the following analysis, only the 7BL locus was considered.

LD analysis based on the r² measure between significant SNPs (*p* < 0.001) demonstrated that tightly linked markers within a QTL on the long arm of chromosome 7B were located within a 658,219,587–680,161,741 bp interval on the physical map. Excalibur_c31707_302 was distant from other SNPs (about 21 Mb), however, it was found to be tightly linked genetically (r² ~ 0.9–1). According to GWAS, the 7B locus negatively influenced the trait's manifestation by reducing MT, whereas markers Kukri_c31628_571, Tdurum_contig43995_611, Excalibur_c31707_302 demonstrated the highest effect hastening the maturity by about 9 days (Table 1).

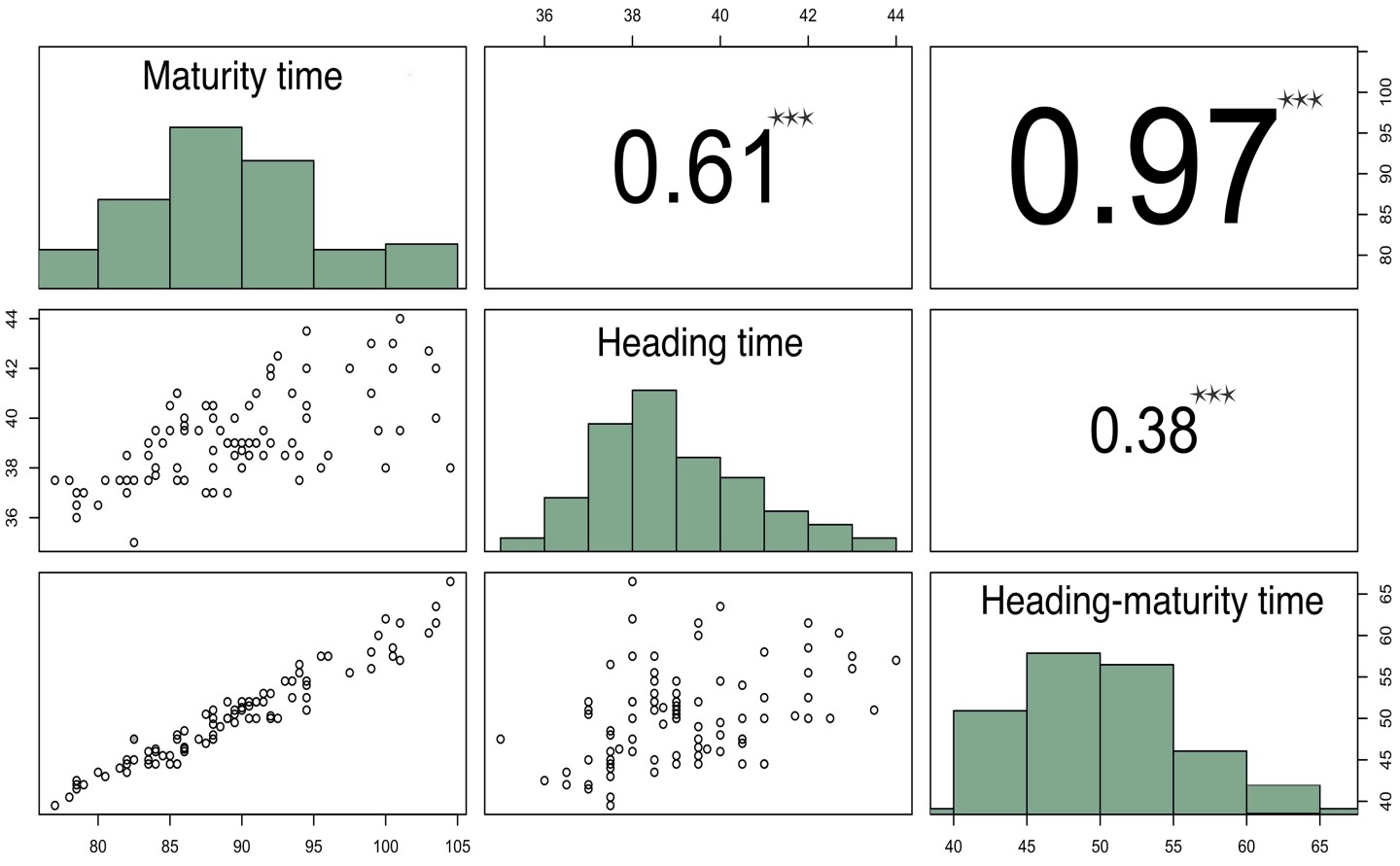

**Figure 1 Correlation matrix plot indicating the significance levels between the estimated developmental periods.** The lower triangular matrix is composed by the bivariate scatter plots. The upper triangular matrix shows the Spearman correlation plus significance level (as stars). Each significance level is associated to a symbol: *p*-values 0.001 (\*\*\*).

### Association of NAM-A1, Vrn-B3 and Vrn-B1 alleles with maturity time

In the course of a grain protein content study performed earlier, we had genotyped the *NAM-A1* gene in the studied panel of wheat cultivars (*Leonova et al., 2022*). In this study, we compared the MT estimated for 11 years and *NAM-A1* alleles. The results indicated that the *NAM-A1* allele variation had no significant impact of on the MT (Fig. S1).

Allelic variation at the *Vrn-B3*, *Ppd-D1*, *Vrn-A1*, *Vrn-B1* and *Vrn-D1* loci was determined by *Berezhnaya et al. (2021)*. The authors observed almost no variation at the *Vrn-A1*, *Vrn-D1*, and *Ppd-D1* loci. Since *Vrn-B1* and *Vrn-B3* are polymorphic, their MT association was assessed. The results de/monstrated that the *Vrn-B1* and *Vrn-B3* alleles or their combination had no significant effect on the MT (Fig. S2).

### Detection and prioritization of candidate genes

Significant SNPs spanned 21.9 Mb region of 7BL that included 216 genes. We estimated the functional annotation of these genes and their expression profiles in different tissues and developmental stages to select mostly appropriate candidate genes.

When evaluating the candidates' expression patterns, we relied on information about the expression of known MT-associated genes. *NAM1* are the key genes determining MT

**Table 1 Marked-Trait associated SNP markers detected from 11-year phenotyping (from 2005 to 2014, 2018) of 92 spring wheat varieties, cultivated in the two experimental fields of ICG SB RAS in the Novosibirsk Region.**

| Marker | Chr | Position | F | $p$ | $R^2$ | MAF | Allele | Allele effect |
|---|---|---|---|---|---|---|---|---|
| Tdurum_contig47508_250 | 2A | 754,339,135 | 8.01 | 7.17E−04 | 0.14 | 0.12 | C | 2.49 |
| BS00023075_51 | 2A | 754,339,135 | 7.97 | 7.46E−04 | 0.14 | 0.12 | G | 3.10 |
| Kukri_c7804_2504 | 3B | 457,025,568 | 8.40 | 5.20E−04 | 0.15 | 0.31 | T | 3.59 |
| RAC875_rep_c83245_239 | 3B | 480,343,168 | 7.67 | 9.67E−04 | 0.14 | 0.30 | A | 3.50 |
| RAC875_c43893_213 | 3B | 482,004,501 | 7.85 | 8.34E−04 | 0.14 | 0.30 | G | 3.74 |
| Excalibur_c41752_392 | 3B | 493,650,726 | 8.06 | 6.89E−04 | 0.14 | 0.29 | C | 3.85 |
| BS00057451_51 | 3B | 493,655,318 | 7.80 | 8.57E−04 | 0.15 | 0.30 | C | 4.05 |
| wsnp_Ku_rep_c68565_67614479 | 4A | 594,215,456 | 7.71 | 9.14E−04 | 0.17 | 0.36 | C | −3.78 |
| Excalibur_c53111_215 | 5B | 711,629,370 | 9.24 | 2.64E−04 | 0.18 | 0.11 | A | −5.37 |
| Tdurum_contig13245_119 | 7A | 10,908,207 | 7.88 | 8.18E−04 | 0.14 | 0.35 | T | −1.69 |
| Tdurum_contig43995_370 | 7B | 658,219,587 | 12.48 | 2.32E−05 | 0.20 | 0.07 | C | −4.73 |
| Kukri_c31628_571 | 7B | 658,219,677 | 15.10 | 3.35E−06 | 0.23 | 0.07 | C | −8.95 |
| Tdurum_contig43995_611 | 7B | 658,219,828 | 15.10 | 3.35E−06 | 0.23 | 0.07 | G | −8.95 |
| Tdurum_contig44993_359 | 7B | 662,652,952 | 11.44 | 4.87E−05 | 0.19 | 0.09 | A | −6.64 |
| RAC875_c89312_61 | 7B | 678,832,328 | 9.47 | 2.24E−04 | 0.16 | 0.12 | C | −6.94 |
| IAAV8521 | 7B | 679,794,071 | 9.55 | 2.07E−04 | 0.16 | 0.12 | T | −6.92 |
| Tdurum_contig5360_329 | 7B | 679,794,384 | 9.91 | 1.61E−04 | 0.17 | 0.12 | G | −7.94 |
| IAAV6659 | 7B | 679,794,445 | 10.10 | 1.36E−04 | 0.17 | 0.12 | A | −7.31 |
| GENE-4442_121 | 7B | 679,799,993 | 16.00 | 1.52E−04 | 0.14 | 0.10 | C | −4.27 |
| Excalibur_c31707_302 | 7B | 680,113,513 | 13.99 | 7.19E−06 | 0.22 | 0.08 | T | −8.80 |
| BS00047083_51 | 7B | 680,161,741 | 8.25 | 6.02E−04 | 0.14 | 0.12 | G | −5.67 |
| BS00064905_51 | UN | 314,905,379 | 13.27 | 5.05E−04 | 0.13 | 0.30 | C | −2.79 |

**Note:**
Chr, chromosome; Position, position of marker on chromosome bp; F, F-test for marker; $p$, $p$-values for marker; $R^2$, marker $R^2$ (%) for maturity time; MAF, minor allele frequency.

duration. The wild-type *Nam-B1* allele has had a negative effect on yield traits, and it is rare in wheat cultivars (*Hagenblad et al., 2012*). Therefore, we couldn't assess its expression pattern. The gene's homoeologues, *Nam-A1*, and *Nam-D1*, are predominantly expressed in the flag leaf after the beginning of flowering, as reported on the website http://www.wheat-expression.com/.

To select the genes whose expression was specific to the flag leaf, the data obtained by *Ramírez-González et al. (2018)* were analyzed. The expression patterns of 216 genes in different tissues were assessed and the genes with preferable expression patterns were selected (File S1). There were twelve genes expressed in the flag leaf: *TraesCS7B02G391800, TraesCS7B02G395600, TraesCS7B02G396600, TraesCS7B02G398700, TraesCS7B02G399700, TraesCS7B02G400500, TraesCS7B02G400600, TraesCS7B02G401600, TraesCS7B02G404300, TraesCS7B02G411800, TraesCS7B02G412200, TraesCS7B02G413400* (Table S5).

## Expression estimation of candidate genes

To select the best reference genes for the plant developmental stages and conditions, the expression of seven previously published reference genes was evaluated. Analysis of the

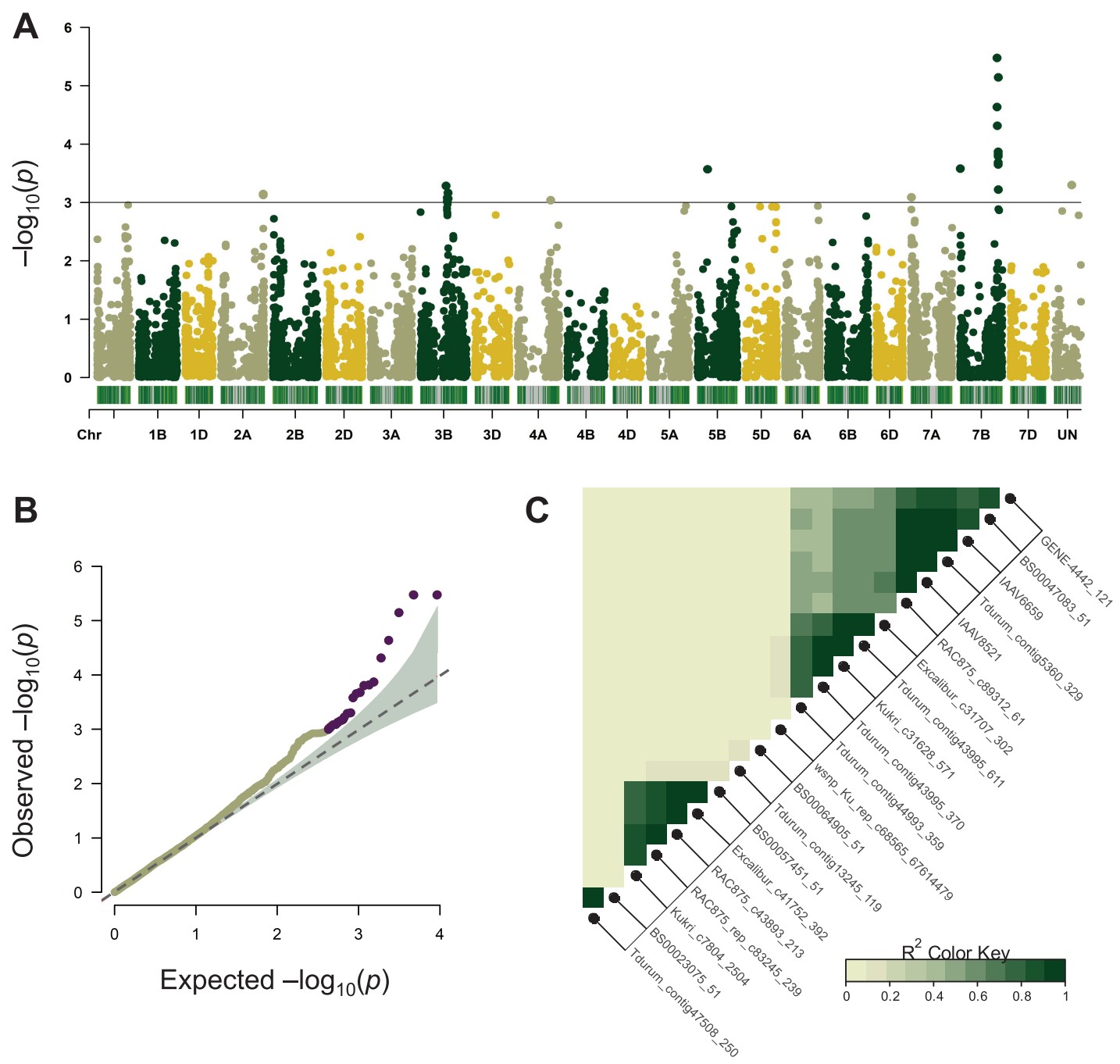

**Figure 2 Summary of genome-wide association studies (GWAS) for maturity time.** (A) Manhattan plots based on the linear mixed model using kinship and Q matrices. The bar under every chromosome corresponds to marker density. (B) Quantile–quantile plots representing the expected *vs* the observed log$_{10}$ *p*-values. (C) Linkage disequilibrium plot representing the association of significant SNP markers. The designations inside the blocks represent the chromosomes corresponding to SNP markers. The color key (bottom right) indicates LD strength from 0 to 1 depending on the color density.

expression stability with the geNorm algorithm demonstrated that the best reference genes for wheat in anthesis phase were the genes encoding scaffold-associated regions (SAR) binding protein and methionine aminopeptidase 1 (MetAP 1) (Fig. S3).

To estimate the candidate gene expression, the most early- and late-maturing varieties having respective haplotypes CCGACTGACT and TAAGTAGTC within the QTL on 7B were analyzed. It was found that eight candidate genes *TraesCS7B02G411800*, *TraesCS7B02G401600*, *TraesCS7B02G400600*, *TraesCS7B02G399700 TraesCS7B02G404300*, *TraesCS7B02G400500*, *TraesCS7B02G398700*, *TraesCS7B02G413400* were not expressed. Expression of three genes (*TraesCS7B02G391800*, *TraesCS7B02G395600*, *TraesCS7B02G396600)* was higher in early-maturing cultivars, but no significant difference in their expression level was found. On the contrary, the expression of *TraesCS7B02G412200* was significantly lower in the early samples ($p < 0.01$) (Fig. 3).

### Validation of association analysis results

Evaluation of the mapped population (Obskaya 2 × Novosibirskaya 15) showed that the offspring MT varied from 74.5 to 91.2 days. The SNP markers mapped on chromosomes 2A, 4A, 5B, and 7B (Table 1) were polymorphic between the parental varieties and the $F_{2:3}$ progeny, while the markers located on chromosomes 3B and 7A were found to be non-polymorphic according to the genotyping performed with an Illumina 15 K Wheat array. There was a significant difference in MT for wsnp_Ku_rep_c68565_67614479 and Excalibur_c53111_215 localized on chromosomes 4A and 5B, respectively (Fig. 4). No significant MT difference was shown found for SNP markers Tdurum_contig47508_250 and BS00023075_51 (chromosome 2A).

The markers on chromosome 7B were divided into two groups according to their genetic linkage and physical map location. Group I included four markers (Tdurum_contig43995_370, Kukri_c31628_571, Tdurum_contig43995_611, Tdurum_contig44993_359) and two haplotypes (TAAG and CCGA). Group II included seven markers (RAC875_c89312_61, IAAV8521, Tdurum_contig5360_329, IAAV6659, GENE-4442_121, Excalibur_c31707_302, BS00047083_51) and two haplotypes (TGAGTCA and CTGACTG). Comparison of the $F_{2:3}$ progeny showed that the TAAG and TGAGTCA genotypes matured later compared to the CCGA and CTGACTG ones (Fig. 4). These MT-related differences had a pronounced trend, despite the lack of significance. It should also be noted that in all cases, the alleles associated with later maturation were inherited from cv. Obskaya 2.

## DISCUSSION

MT includes two main stages: HT and HTMT. The genetic basis of HT in wheat has been studied quite well. Although many components of this pathway are still unknown, the key factors involved in the transition of a plant from the vegetative to the generative developmental phase such as vernalization genes (*VRN1* and *VRN2*), photoperiod sensitivity genes (*PPD1*), light signal perception genes (*PHY*), many circadian rhythm genes, some microRNAs, and an integrator of most known pathways *Flowering Locus T* (*VRN3, TaFT1*) have been studied in detail.

Most studies address the first phase, namely HT, while the genetic component of the maturity time in wheat and other cereals remains poorly understood. In our study, we demonstrated that in the long day, typical for Western Siberia, MT is markedly affected by

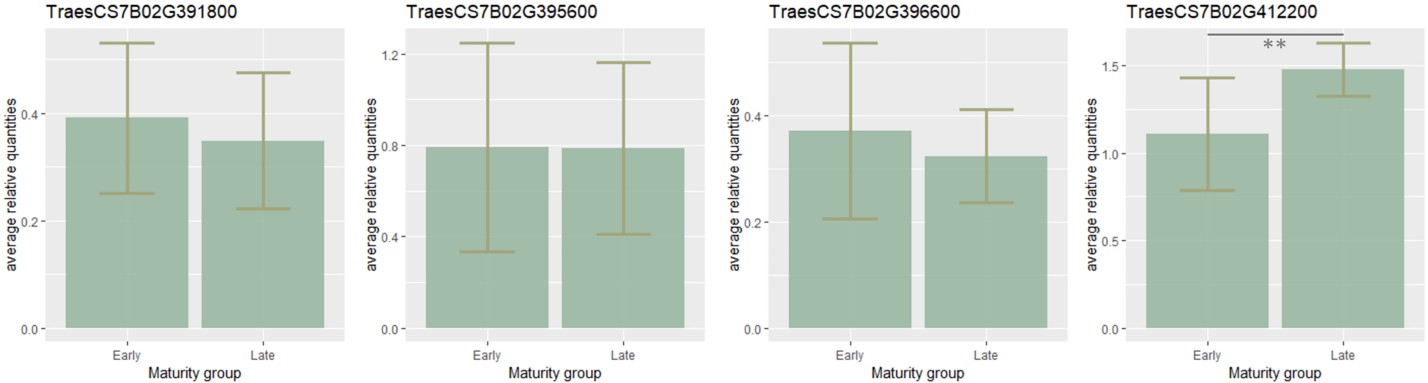

**Figure 3 Barplots of candidate-gene relative expressions in early- and late-maturing wheat varieties.** Error bars indicate standard deviation (SD). The asterisks indicate significant differences (**$p < 0.01$).

HTMT but not by HT itself. Previously, many authors had noticed that MT did not always depend on HT. For example, *May & Van Sanford (1992)* had shown that HT was not always highly correlated with MT. However, no specific genes had been defined for MT in wheat. Thus, investigating a genetic basis for HT only, reduces the number of sources for breeders to create highly adapted varieties.

There is inconsistent information about heritability rate of wheat maturity. Thus, *Hossain et al. (2021)* has indicated that maturity days of spring wheat genotypes have low heritability (35%) with low genetic advance. In another work, *Mohsin, Khan & Naqvi (2009)* have shown, that wheat physiological maturity has moderate heritability value (63%) and genetic advance. Our results agreed with the data of *Ullah et al. (2011)* and those of *Al-Tabbal & Al-Fraihat (2012)*, who have reported that maturity time is highly heritable trait (82% and 94%, respectively).

It can be supposed that such differences in the data on trait heritability are due to the diversity of the environmental condition in the regions where investigations were performed, and the characteristics of the studied genotypes. Our experiment, considering a set of the spring common wheat varieties cultivated in conditions of Western Siberia, showed a high level of heritability (73.5%) for the MT trait, so a significant contribution of the genetic component to the formation of the trait can be assumed, which was confirmed by ANOVA demonstrating a significant impact the genetic component and environment had on the maturity time. Hence, selection for this trait in our case makes sense, and the identification of the new loci and genes associated with it can improve the varieties' adaptation.

## Association of the known genes with maturity time

To date, *NAM1* are the only genes whose influence on the time of wheat ripening has been confirmed. It is known that the wild-type allele of the *NAM-B1* gene is associated with earlier maturity, while it is almost not found in modern cultivars (*Hagenblad et al., 2012*). Also, the *NAM-A1a* and *NAM-A1b* alleles are specific for early and medium-early varieties (*Alhabbar et al., 2018*).

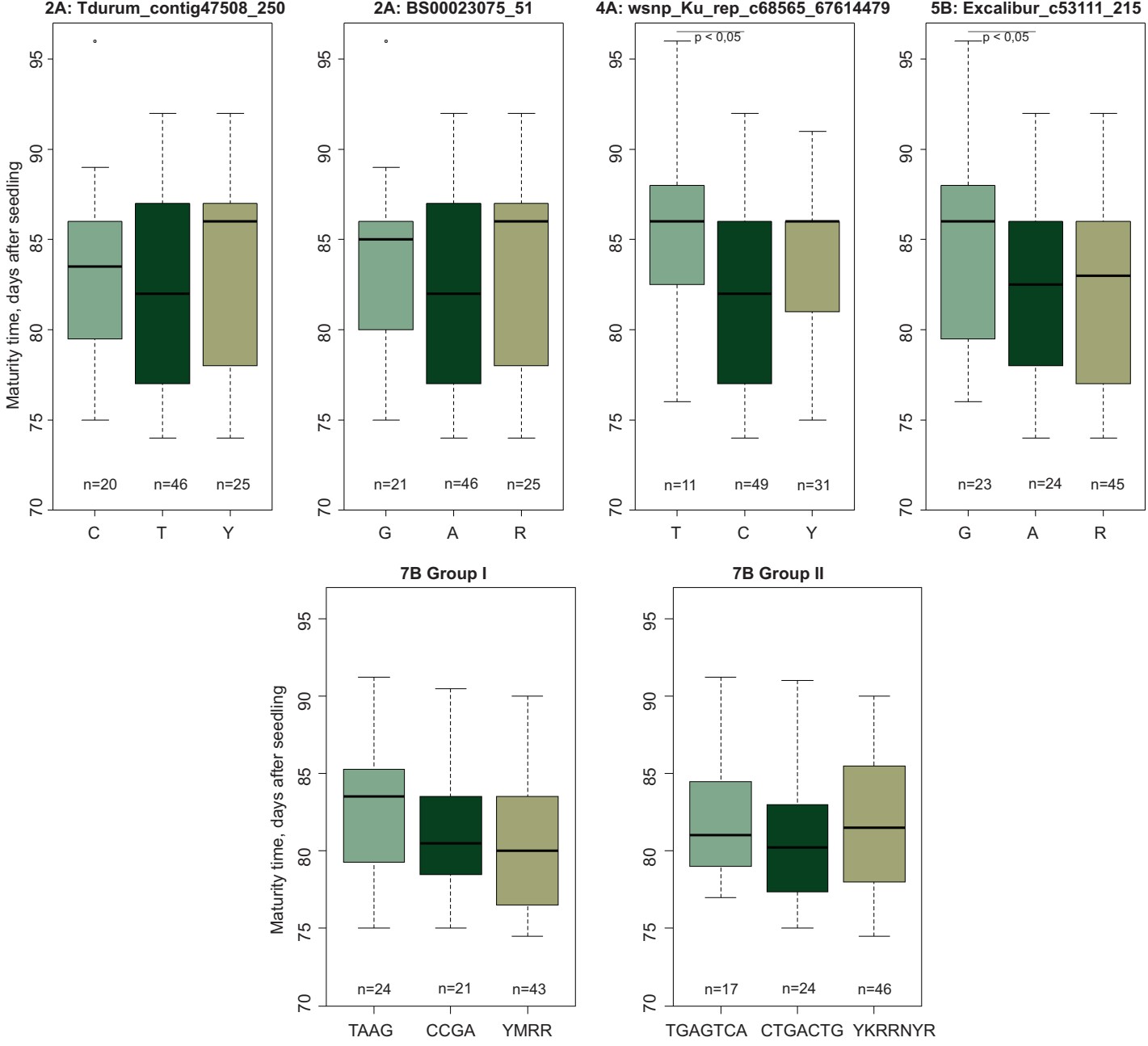

**Figure 4 Boxplots of maturity time variation in groups of genotypes with different SNPs.** YWRR and YKRRNYR are designation of hetero-zygous genotypes. "*n*" denotes the number of samples in each group.

In previous study, analysis of the spring common wheat panel identified the *NAM-A1a*, *NAM-A1c*, and *NAM-A1d* alleles (*Leonova et al., 2022*). Since in our sample only five varieties carried the *NAM-A1a* allele, which together with *NAM-A1b*, is associated with early maturity, it is no wonder that no significant relationship between the allelic state of the *NAM-A1* gene and MT was detected.

*Vernalisation* and *Photoperiod* genes are key factors of wheat development and some works have demonstrated that they may determine not only heading and flowering times

but maturity time as well (*Shcherban, Emtseva & Efremova, 2012*; *Kamran et al., 2013*; *Zaitseva & Lemesh, 2015*; *Whittal et al., 2018*).

The wheat samples in our study were analyzed to identify the alleles of the *Vernalisation* genes. The plants had different alleles of *Vrn-B1* and *Vrn-B3* genes, while *Vrn-A1*, *Vrn-D1*, and *Ppd-D1* were not found to be polymorphic. However, we did not observe any significant impact of *Vrn-B1* and *Vrn-B3* on the heading time of the examined wheat plants.

Among the *VRN-1* gene alleles, accelerated development was predominantly associated with the *Vrn-A1a* allele, which was present in the majority of the sampled varieties. Similarly, the dominant alleles of *Vrn-B1* (*Vrn-B1a* and *Vrn-B1c*) detected in our panel of cultivars were unlikely to have a different effect on development.

In our investigation of the *Vrn-B3* gene, we identified two alleles: *vrn-B3*, which is the recessive variant typically found in late cultivars, and the *Vrn-B3e* allele, which delays heading time by 1.5 days when compared to *vrn-B3*.

The ANOVA results didn't demonstrate a significant impact of the Vrn-B3 alleles on the maturity time. We also determined that under the conditions of the experiment, the main contribution to maturity time inputed the period from heading to maturity. Based on this finding, it can be suggested that *Vrn-B3e* does not appear to alter MT (maturity time) in spring wheat cultivars.

## QTL colocalization with previous studies

In this study significant MT-associated SNPs were found on chromosomes 2A, 3B, 4A, 7A, and 7B. The most significant markers were located on the 7B chromosome. Previously, *Kulwal et al. (2003)* had reported about a QTL on the long arm of the 7B chromosome to be associated with MT. The QTL was tightly linked with RFLP marker XksuD2 located on 7BL (*Paillard et al., 2003*). The Blast analysis of the XksuD2 primers specified that it was located in the 683,517,074–683,517,093 region of 7BL, a position to be very close to the MTAs detected on 7BL in our study.

The second locus coincided with the previously detected region was positioned on the short arm of the 7A chromosome. Its MTA was associated with SNP marker Tdurum_contig20214_279 positioned in 61,816,447. The location of QMat.dms-7A.2 (*Perez-Lara et al., 2016*) was associated with RAC875_c14982_577 (position 59,045,900). So, these two loci may be assumed to be very close to each other and should be the same. However, MT-associated loci had been previously detected on the same chromosomes: 2A (*Wang et al., 2009*), 3B (*McCartney et al., 2005*), 4A (*McCartney et al., 2005*; *Chen et al., 2015*; *Perez-Lara et al., 2016*), 7A (*Zou et al., 2017*). Comparative analysis of their location demonstrated that they were different.

## Candidate genes

*NAM1*, a maturity time gene, is expressed after flowering begins in the flag leaf. Moreover, a number of studies have demonstrated the flag leaf to be main source of assimilates during the grain-filling period, which concurs with senescence onset (*Ashraf & Bashir, 2003*;

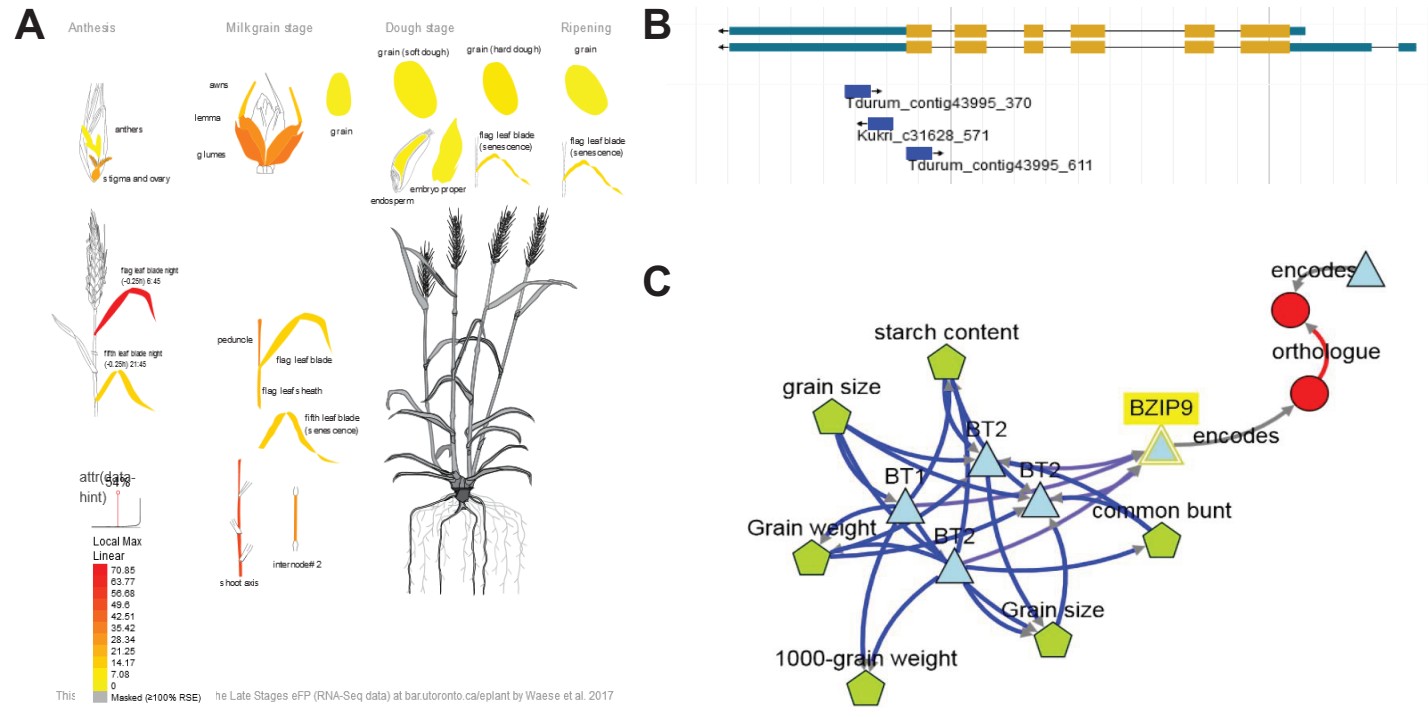

**Figure 5 Summary information about candidate gene *TraesCS7B02G391800*.** (A) Expression patterns of *TraesCS7B02G391800* in different tissues and developmental stages visualized using the Wheat efpBrowser (https://bar.utoronto.ca/efp_wheat/), which utilizes expression data published in *Ramírez-González et al. (2018)* and *Winter et al. (2007)*. EfpBrowser, which is a part of BAR (The Bio-Analytic Resource for Plant Biology), is under the U.S. Public Domain. (B) Localization of significant SNPs in the *TraesCS7B02G391800* sequence using the GrainGenes genome browser (*Yao et al., 2022*). GrainGenes provide information for free public use. (C) Candidate gene network designed using the KnetMiner (https://knetminer.com/, *Hassani-Pak et al., 2021*). KnetMiner is under the MIT license.               

*Viljevac Vuletić & Španić, 2020*). This is the reason why we focused on the candidate genes in the QTL that were most strongly expressed in the flag leaf.

Annotations of these twelve genes are summarized in Table S5. Most of them are predominantly expressed in flag leaf after the heading or flowering begins.

One of the genes showing strong expression in the flag leaf at the flowering stage was *TraesCS7B02G391800* (Fig. 5A). Three the most significant SNPs (Tdurum_contig43995_370, Kukri_c31628_571, Tdurum_contig43995_611) were linked with *TraesCS7B02G391800* (Fig. 5B). These SNPs are either located in the last exon (Tdurum_contig43995_611) or in 3′UTR region (Tdurum_contig43995_370, Kukri_c31628_571). The gene encodes a BZIP domain-containing protein. BlastP for Arabidopsis sequences demonstrated that the protein encoded by *TraesCS7B02G391800* was related to such bZIP transcription factors as bZIP9, BZO2H1 (bZIP10), bZIP25, bZIP63, and the bZIP9 protein known to regulate bZIP53, which, upon dimerization with bZIP10 or bZIP25 (both C-group bZIPs), may enhance maturity-gene expression (*Alonso et al., 2009*; *Jain, Shah & Rishi, 2018*). *TraesCS7B02G391800* is orthologous to rice *OsbZIP52* (*Os06g0662200*) that encodes a probable transcription factor that binds to the DNA specific sequence 5′-TGAGTCA-3′ found in seed storage protein gene promoters (PubMed:11133985).

However, qPCR revealed no significant difference of *TraesCS7B02G391800* expression in the early and late maturing varieties, we suppose this gene to be a highly probable candidate.

*TraesCS7B02G401600* is expressed in the flag leaf while anthesis and then in the embryo proper while the ripening stage. The gene is known as *BRITTLE1* (*BT1*) and provides unidirectional transmembrane transport of ADP-glucose that is crucial for starch synthesis in cereal grain (*Wang et al., 2019*).

KnetMiner demonstrated that genes *TraesCS7B02G391800 (bZIP9)*, *TraesCS7B02G401600 (BT1/BT2) and TraesCS7B02G413400* may interact each other (Fig. 5C), which allows us to assume that in the identified locus, several genes might affect MT thanks to their interaction with each other.

*TraesCS7B02G412200* (encode photosystem II reaction center) is expressed in the flag leaf after the flag-leaf stage begins. It was the only gene that demonstrated significant expression difference between early- and late-maturing variety groups in our qPCR experiment. Previous studies had demonstrated that the genes for chlorophyll binding, chloroplast, chloroplast organization, photosystem I, and photosystem II were down-regulated in early senescing varieties (*Yoshida, 2003*; *Zhang et al., 2018*).

KnetMiner analysis of the gene networks involving *TraesCS7B02G412200* revealed its relation to *CGA1* (*CYTOKININ-RESPONSIVE GATA TRANSCRIPTION FACTOR*) (Fig. S4). Cytokinins are key regulators of plant growth and development, including senescence (*Argueso, Raines & Kieber, 2010*). And *CGA1* overexpression is known to be associated with delayed senescence and reduced grain filling (*Hudson et al., 2013*). The decay of chloroplasts produces most of the nitrogen reassimilated to the seeds (*Masclaux-Daubresse, Reisdorf-Cren & Orsel, 2008*). *Hudson et al. (2013)* have hypothesized that expression of *CGA1* results in increased activity of chloroplasts, that in turn causes the delayed senescence, inhibits nutrient remobilization into seeds and prevents proper grain filling, which makes it possible to assume that *TraesCS7B02G412200* and the genes associated with it can indirectly influence the maturity time through the regulation of photosynthesis and chloroplast functioning.

## CONCLUSIONS

Thus, based on the results obtained, a conclusion can be made that the most significant QTLs that reduce the duration of the period from germination to maturity are located on chromosomes 4A, 5B, and 7B, while the 7B locus has the most significant contribution to the expression of the trait, shortening the maturity time by almost 9 days. The QTL validation performed using the $F_{2:3}$ population mapping has shown that markers wsnp_Ku_rep_c68565_67614479 (chromosome 4A) and Excalibur_c53111_215 (chromosome 5B) are significantly associated with early maturity. Group I haplotype CCGA and group II haplotype CTGACTG on 7BL have demonstrated a strong tendency to accelerate maturity time. To confirm the contribution of 7B haplotypes to the trait, it is necessary to perform experiment in additional environments. We suppose that the early-maturity QTLs identified on 4A, 5B and 7BL can be used as a valuable source for breeding and could contribute to the genetic improvement of wheat.

### Funding

The study was carried out with financial support by the Russian Science Foundation (RSF project No. 21-76-30003). Validation of association analysis results was performed within the budgetary project FWNR-2022-0017. The funders had no role in study design, data collection and analysis, decision to publish, or preparation of the manuscript.

### Grant Disclosures

The following grant information was disclosed by the authors:
Russian Science Foundation (RSF): 21-76-30003.
Validation of Association Analysis Results was Performed within the Budgetary Project: FWNR-2022-0017.

### Competing Interests

Elena A. Salina is an Academic Editor for PeerJ.

### Author Contributions

- Antonina A. Kiseleva conceived and designed the experiments, performed the experiments, analyzed the data, prepared figures and/or tables, authored or reviewed drafts of the article, and approved the final draft.
- Irina N. Leonova conceived and designed the experiments, performed the experiments, analyzed the data, authored or reviewed drafts of the article, and approved the final draft.
- Elena V. Ageeva performed the experiments, prepared figures and/or tables, and approved the final draft.
- Ivan E. Likhenko performed the experiments, prepared figures and/or tables, and approved the final draft.
- Elena A. Salina conceived and designed the experiments, authored or reviewed drafts of the article, and approved the final draft.

### Data Availability

The raw data for SNP genotyping, phenotyping and full GWAS results are available in the Supplemental Files.

### Supplemental Information

Supplemental information for this article can be found online at http://dx.doi.org/10.7717/peerj.16109#supplemental-information.

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
