# Peer review of "Identification of genetic loci for early maturity in spring bread wheat using the association analysis and gene dissection"

_PeerJ, doi:10.7717/peerj.16109_

## Round 0.1 · original submission · Major Revisions

Dear authors,
Regarding the submitted manuscript “Identification of genetic loci for early maturity in spring bread wheat using the association analysis and gene dissection”.
Two expert reviewers have now completed the assessment of your work and concluded that it is suitable for publication after major revisions.

Sincerely,
Dr Nikolaos Nikoloudakis

Reviewer 1 ·

Basic reporting

This paper contains interesting studies on identification of genetic loci for early maturity in spring bread wheat using the association analysis and gene dissection. Although many studies have been conducted on that is still lacking in English and more informations. However, this manuscript still needs improving in writing logic and the analysis of the discussion . It is recommended to modify for a better version. Moreover the discussion section is not deep enough.

Experimental design

-this part needs to describe very well by using suitable subheadings. However, it needs more of the data and information’s for the investigations, specially regards material process and statisticians. . in the revised version to enhance clarity with standard methodology.

Validity of the findings

-it needs to rearrange with subsections for measurements and figures needs further improvements to be more clear and understanding .

Additional comments

This paper contains interesting studies on identification of genetic loci for early maturity in spring bread wheat using the association analysis and gene dissection. Although many studies have been conducted on that is still lacking in English and more informations. However, this manuscript still needs improving in writing logic and the analysis of the discussion . It is recommended to modify for a better version. Moreover the discussion section is not deep enough.

Title: It needs to modify .
Abstract: it should consider focusing the important findings briefly .
Keywords:should be re- consider and updating.

Abbreviations: should arrange

Introduction:
-Introduction part is not appropriate ,but, it needs for further improvements with highlighting research gaps which necessitated conducting this experiment for the last 5 years. ESSENTIAL.

Materials and methods:
-this part needs to describe very well by using suitable subheadings. However, it needs more of the data and information’s for the investigations, specially regards material process and statisticians. . in the revised version to enhance clarity with standard methodology.

Results
-it needs to rearrange with subsections for measurements and figures needs further improvements to be more clear and understanding .
Figures must be more clear.
Discussion
-This part needs major revision with providing last 5 years with similar investigations .
Conclusion:
- arrange this part with briefly and respect to formulated objectives.
References:
-Cross check the references in the text and reference cite. Few references are not as per journal style in the text as well reference section.

Reviewer 2 ·

Basic reporting

Clear and Concise:
Overall, the manuscript is well written and the authors should be commended. However, there are a few places that minor grammar issues make the manuscript more difficult to read. A careful, thorough read is suggested. (line 28 "not by a heading time", line 108 "for the same spring wheat variety panel", line 153, "normality distribution", line 159 "SNPs", line 272 incomplete thought, line 277 "expressing in the flag leaf" -> expressed in the flag leaf ).

Literature and References:
Overall the literature references are appropriate. I would suggest a better connection of paragraph (Line 79-93) to wheat literature. Specifically, are there homologs of these genes in wheat? Especially, with the claim made in line 95 "in wheat have not yet been identified," it seems like maybe they haven't been characterized or validated in wheat. Also, many inline references need authors removed from citation i.e. line 337 Al-Tabbal and Al_fraihat (Al-Tabbal and Al_Fraihat, 2012)

Line 72: It seems like this should only be functional NAM-B1, as previous line 70-71 show that non-functional (which should include deletions) resulted in late maturity.

Professional Strcuture:
Content of figures and tables are appropriate. Table1: Add Marker-trait, could add more description where and when the 11 year phenotyping trials were. Is Tdurum_contig47508_250 and BS00023075_51 at the exact same position?
Table S1 and S2 headings should probably include the species of study--wheat, and possibly what lines--a spring wheat panel for maturity analysis included. Table S2 could include where primer sequences originated from--(i.e. BLAST search, literature, etc.)

Table S3. Needs more descriptive heading. From the table we get time, but a description of what (wheat and maturity time), where (Western Siberia), why (panel for maturity time analysis). Footnotes describing SE, CI, std, coef would be nice--then possibilities of misinterpretation would be greatly decreased.

Table S4. Same as S3, please add more descriptive heading.

Figure S2 & S3: Does NAM_A1a have whisker lines? Maybe add number of lines of each allele (n) below boxplots so readers know how big a group was analyzed at each haplotype.

Figure S2 and S3 are listed before Figure S1

Self-contained relevant results:
Results are relevant and appropriate, though at times possible far reaching for the data (see section Experimental Design). Particularly the conclusions (Line 409-410) should be double checked. Unsure how if qPCR showed no relationship it is still a probably candidate? Is this based on sample size, limits of detection, etc.?

Experimental design

Original primary research: (Good)

Well defined research: (Good) genetic dissection of maturity time in spring wheat in Western Siberia.

Rigorous Investigation: (Excellent)

Methods described in sufficient detail:
I would suggest some areas to clarify including:
Line 227: Was there a specific genotype-by-environment term added in the model?

Line 118: For complete clarity, is number of days from seedling to heading, the days after seedling emergence, which is then calculated as? For example, were 50% of plot expected to be emerged to be seedling day 1? Same type question in line 212: for total clarity is the time in days from seedling emergence to maturity. Line 217: 35-44 days is after seedling emergence? Similar to line 218 77-104 days. As the manuscript is about heading and maturity time, a very concise and strict usage of these words is recommended throughout the manuscript.

Line 187: (three bios for every cultivar), just to clarify this is three plants pooled together for each of the 10 cultivar--10 total samples? or 3 plants of each cultivar into one sample replicated 3 times for a total of 30 plants sampled?

Line 137-140: Is sum of effective temperatures, the equivalent of growing degree days (GDD)? GDD is quite common in literature, and I would suggest this method rather than effective temperature.

Line 363-371: Possible unclear, difficult to tell how Vrn-B3e does not effect MT. It appears that HD is delayed so if MT is the same length it would shorten MT. Possible revise for clarity.

Line 111: "maturity time from seedling to maturity" Is the first maturity the same as development time?

Line 154: Were other traits for heritability also estimated?

Validity of the findings

Meaningful replication:
I find this to be possible the weakest part of the manuscript--at no fault to the authors; however, the number of each haplotype/allele of associated MTA's should be included. While it is somewhat included, (line 355), I think full utilization of the results necessitate being more direct that some sample sizes are quite small. Expanding and making this very explicit, aids the reader in understanding that while the data point to these results, smaller sample size for some values may impact future performance. As the authors conclude (Line 445) "additional experiments ..." are needed to verify these results.


All data provided: (excellent)

Conclusions: (good)

---

## Round 0.2 · accepted · Accept

Dear colleagues/PeerJ staff,

I have carefully read the revised manuscript, as well as the authors' rebuttal letter.

It is my opinion that the manuscript has been significantly improved based on the comments raised by reviewer No2 and I have concluded that the authors have successfully included/addressed them.

Reviewer No. 1 raised some very unclear concerns, and the authors did their best to justify them.

As a result, I believe that the manuscript can be accepted as it stands.

Reviewer 2 ·

Basic reporting

no comment

Experimental design

no comment

Validity of the findings

no comment

Additional comments

The authors have made a good faith effort to address all review criteria. I believe the manuscript has been greatly improved based on the author changes to reviewer comments.